



# Rill Erosion on Slope of Spoil tips: experimental study of
# runoff scouring erosion in multiple times
Yongcai Lou[1], Zhaoliang Gao[1,2], Fuyu Zhou[1], Jianwei Ai[1], Yunfeng Cen[1], Tong Wu[2]
and Jianbin Xie[3]
[1]State Key Laboratory of Soil Erosion and Dryland Farming on the Loess Plateau,
Institute of Water and Soil Conservation, Northwest A&F University,Yangling,Shaanxi,
China.
[2]Institute of Soil and Water Conservation, Chinese Academy of Sciences and Ministry
of Water Resources, Yangling, Shaanxi, China.
[3]College of Architecture and Plianning, Kunming, Yunnan, China.
*Correspondence to*:Zhaoliang Gao (gzl@ms.iswc.ac.cn)
**Abstract.** The soil erosion of the spoil tips seriously threatens the safety of people's lives and property
and the surrounding ecological environment. Rill erosion is an important cause of water and soil loss in
spoil tips. This study was conducted to investigate the process of rill erosion on the slopes of spoil tips,
changes in the morphological characteristics of rills and the mechanisms of rill erosion. A Field runoff
plot (5 m long, 1 m wide and 0.5 m deep) with three inflow rates (1.6, 2 and 2.4 mm min$^{-1}$) and three
typical slopes (28°, 32° and 36°) was used for runoff simulation experiments. The results showed that,
compared with the slope and scouring times, inflow rate was the most important factor affecting rill
erosion of the spoil tips. The development of rill mainly goes through three stages: the rill formation
stage, the rill development stage and the rill adjustment stage. The overall predominance of parallel-
shaped rills at all experiments suggested that the formation of rills was dominated by concentrated runoff.
The average rill depth was the best indicator of rill morphology for evaluating rill erosion. The flow
regimes under the experimental conditions were supercritical-laminar flow and supercritical-transition
flow. The Reynolds number was the best hydraulic parameter for predicting rill erosion. The stream
power was the best hydrodynamic parameter to describe rill erosion mechanism. These results
contributed to further revealing the rill erosion mechanism on the slope of the spoil tips and provided a
scientific basis for its soil erosion control.
**1 Introduction**
Transportation, water conservancy, mining and other infrastructure construction industries are
developing rapidly globally, especially in China. As a result, a large amount of spoil tips has been





produced(Niu et al., 2019; Yang et al., 2019). Compared with the undisturbed landscape, the typical
characteristics of spoil tips include loose structure without vegetation-covered, slope with steep gradients
and short length(Zhang et al., 2015; Lv et al., 2019). Its soil erosion rate and erosion intensity far exceed
those of the original landform(Mcclintock and Harbor 2013), causing significantly greater soil loss than
that of eroded landform units such as sloping land and forest land(Kaufman 2000). Previous studies
showed that spoil tips have become a major source of soil erosion from production and construction
projects(Peng et al., 2014). Under the effect of rainfall and runoff, spoil tips are prone to serious
secondary hazards such as soil erosion(Guo et al., 2020), landslides and debris flows(Conforti and Ietto
2020), affecting soil and water resources(Fransen et al., 2001), and the surrounding environment(Owens
et al., 2005), downstream rivers and water and sediment(Morokong and Blignaut 2019). Therefore, it is
necessary to study the processes and mechanisms of erosion of spoil tips.

Sheet erosion, rill erosion, gully erosion and in-stream erosion are the main types of erosion on

slopes(Merritt et al., 2003; Sun et al., 2013). Once rills are formed on the slope, the surface flow will
quickly become concentrated flow. The concentrated flow with fast velocity and strong shear force has
a much greater capacity to detach and transport soil particles than the erosive force caused by rainfall,
which will result in a sudden increase in the amount of erosion on the slope(Auerswald et al., 2009).
Therefore, rill erosion is the most severe erosion form among water erosion on slope, and its occurrence
often marks the gradual development of soil erosion into gully erosion(Chen et al., 2013). Previous
studies have shown that rill erosion is one of the main causes of soil loss and accounts for 70-97 % of
total soil erosion(Zheng and Tang 1997; Whiting et al., 2001; Sun et al., 2013). There are four stages in
the formation of rill: sheetflow, flowline development, micro-rills and micro-rills with head-cuts(Merritt
1984).Understanding of the rill erosion processes on slopes is important not only for the prevention of
soil erosion in spoil tips, but also for soil erosion prediction models.

After the appearance of rills, as the rainfall or scouring continued, the rills bifurcated, merged and

connected on the slope to form a complex erosion pattern that evolves into a crisscross network of
rills(Shen et al., 2015). Rill length, width, depth and related derived indicators (e.g., rill density, rill
complexity, rill width-to-depth ratio)(Cerdan et al., 2002; Tian et al., 2017; Zhang et al., 2017; Qin et al.,
2018) are often used to describe rill morphology. For example, Shen et al. (2019) indicated that the rill
width-depth ratio was a better indicator for analyzing differences in the rill characteristics for treatments
with different slope gradients and for assessing the rill cross-sectional features. Shen et al. (2015)





concluded that the average rill width was the best basic morphological indicator for evaluating rill erosion.
Gilley et al. (1990) suggested that the rill density was a good description of the degree of development
of rills. In the process of rill erosion, the rill morphology is largely determined by the hydrodynamic
characteristics of the rill flow. In addition, rill flow hydraulic parameters (e.g., flow velocity, flow depth,
Reynolds number, Froude number and Darcy-Weisbach coefficient)(Govers et al., 2007; Niu et al., 2019;
Omidvar et al., 2019; Yang et al., 2020) and dynamic parameters (e.g., shear stress, stream power and
unit stream power)(Zheng et al., 2004; Li et al., 2016; Guo et al., 2018) are also often used to describe
the rill erosion mechanism on slopes. For example, by studying the hydrodynamic characteristics of rill
erosion, Nearing et al. (1997), Reichert and Norton (2013) and Shen et al. (2016) found that stream power
can more accurately to characterize the dynamic mechanisms of rill erosion. However, Tian et al. (2017)
showed that shear stress is the best hydrodynamic parameter to describe rill erosion under scouring
conditions. Rill morphology is the result of the interaction between the hydrodynamic factors of runoff
and the soil(Zhang et al., 2015). The development of soil erosion changes the morphology of the rill bed,
which in turn affects the hydrodynamic and erodibility of runoff, and changes in runoff energy led to
further changes in rill morphology(Chen et al., 2015; Xu et al., 2017). The evolution of the rill
morphology, the hydrodynamic properties of runoff and soil erosion thus form a complex mutual
feedback process(Favis-Mortlock 1998; Gatto 2000). Therefore, it is necessary to study runoff hydraulic
characteristics and dynamic mechanism of rill erosion of spoil tips.
Under natural conditions, it has been observed that rills on the slope of spoil tips may be formed by
multiple times rainfall or runoff from upslope. Qin et al. (2018) showed that rill networks evolved in a
converging way, a large number of small rills were formed during the first rainfall, rills were gradually
connected during the second rainfall, the rill network was basically formed, and the rill erosion was
intensified through the process of rill bifurcation, connection and merging, the rill network was further
developed during the third rainfall, and by the fourth rainfall the rill network was mature. However, many
studies have focused on the changes in the rill erosion process of slope during a single rainfall or scouring
process(He et al., 2017; Jiang et al., 2018; Niu et al., 2020; Tian et al., 2020). The impact of multiple
events on rill erosion has been ignored. Therefore, it is necessary to study the rill erosion on the slope of
spoil tips under the multiple times rainfall or scouring conditions.
Studying the rill development and morphological characteristics is of great significance to revealing
the nature of soil erosion on slope of spoil tips, and also provides a theoretical basis for the development



of erosion prediction models. In this study, field experiment was conducted with the objectives of: 1)
analyzing the change process of runoff and sediment yield on slope of spoil tips and quantifying the
effect of slope, inflow rate and scouring times on runoff and sediment, 2) quantifying the changes in rill
networks and morphological characteristics and elucidating the relationship between rill morphological
parameters and rill erosion, and 3) exploring the hydrodynamic mechanism of rill erosion and determine
the best hydrodynamic parameters for predicting rill erosion.
**2 Materials and Methods**
**2.1 Experimental site and soil samples**
The experimental site is located at Yangling Ling Hou Experimental Station
(34°19′24″N,107°59′36″E) (Fig.1a), Institute of Soil and Water Conservation, Ministry of Water
Resources, Chinese Academy of Sciences. The experimental station has a continental monsoon climate,
with an average annual temperature and precipitation of 13°C and 610 mm, respectively, of which more
than 80 % is of short-duration and high-intensity and concentrated in July to September. The runoff plots
are built on hand-excavated side slopes, 20 m long and 5 m wide, with slopes of 28°, 32° and 36°
respectively (Fig.1b).
The experimental soil was obtained from the excavation of the extension project of the experimental
station. The soil used in this experiment was clay loam according to the International Soil Texture
Classification with 28.72 % sand (20 μm–2 mm), 40.12 % silt (20–2 μm), and 31.15 % clay (< 2 μm).

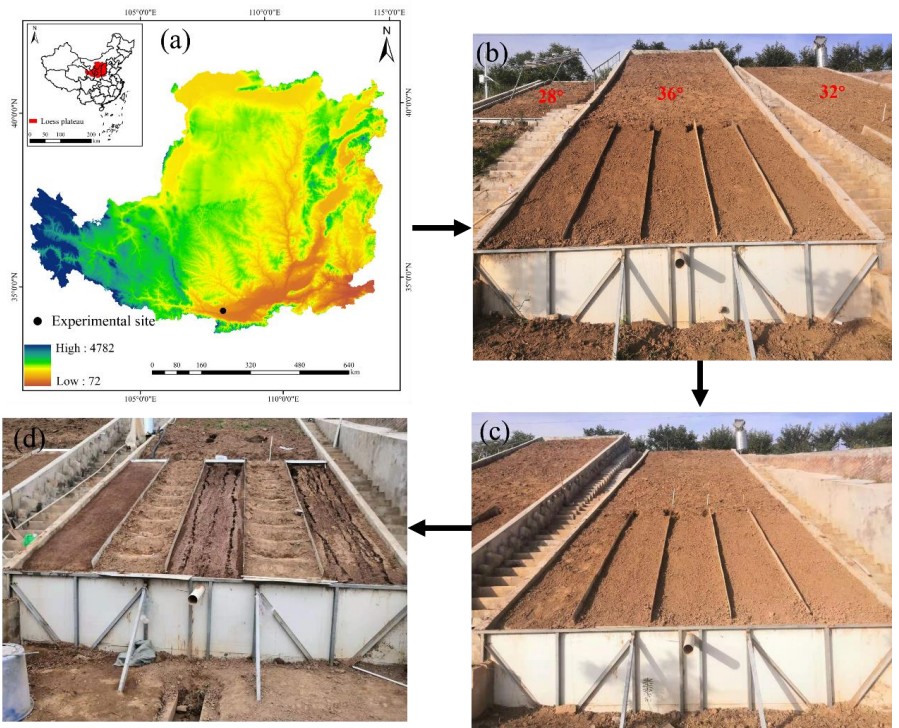


**Figure 1.Location of experimental site(a) , runoff plots (b)with 28°, 32° and 36°and layout of the experimental**

**treatments (c, d)**

## 2.2 Experimental design

Spoil tips, as a special type of artificial landform, consists of a platform and a steep slope (Fig. 2), the platform being the main area where runoff collects and the slope being the main source of eroded sediment(Zhang et al., 2016). Therefore, this paper uses field scouring experiments to simulate the rill erosion of slopes by collected runoff from platforms. The field scouring experimental setup included a water supply line, a constant barrel, a valve, a flow meter, a steady flow groove, and collecting barrels (Fig. 3). In this study, two replicates with three slope gradients, each series contained three successive scouring were applied. Based on the short-duration and high-intensity erosive rainfall criteria ($I_5 = 1.52$ mm min$^{-1}$, $I_{10} = 1.05$ mm min$^{-1}$) for the Loess Plateau, inflow rates of 1.6, 2 and 2.4 mm min$^{-1}$ were applied. The results of the field survey of 368 spoil tips show that the length of slope of 2 to 8 m account for 78.4% of the total survey, with the slope mainly concentrated at 25° to 40°(Li et al., 2020). Therefore, in the experiment runoff plots were divided into 5m lengh×1m width ×0.5m depth with slopes of 28°, 32° and 36° using PVC sheets (Figs. 1c, 1d).


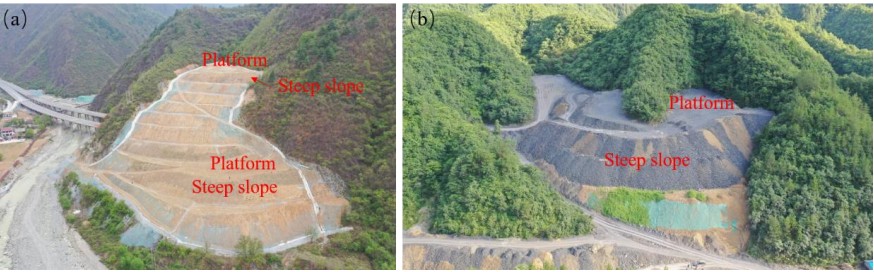


**Figure 2 Spoil tips from highway construction in China**

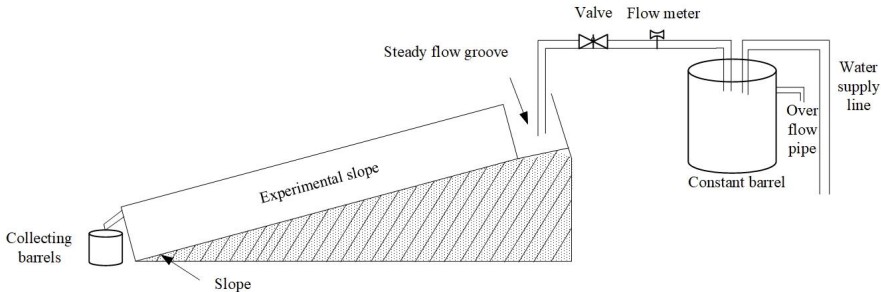

**Figure 3 Layout of the field scouring experimental setup.**
**2.3 Experimental preparation and procedure**
The runoff plots were filled with soil by the layered filling method. Firstly, the bottom layer of the
plots was 5 cm thick with a soil bulk density of 1.45 g cm$^{-3}$, and then, a 25 cm-thick layer of lightly
disturbed soil with a bulk density of 1.32 g cm$^{-3}$, and the top layer is a 20cm-thick heavily disturbed soil
with a bulk density of 1.25 g cm$^{-3}$ to represent typical spoil tips in the region. It should be noted that in
ensuring the natural state of the soil, the experimental soil was filled into the plots without sieving only
to remove plant roots, dead leaves and larger clods of soil(Niu et al., 2020).
Before start of the experiment water was sprinkled evenly on the slope to keep the initial moisture
content consistent. After that, the plots were covered with plastic sheeting and left for 24 h to allow free
infiltration of water close to the natural state of soil moisture distribution. The initial moisture content of
the soil was 13 %.
After the experiment started, runoff and sediment samples were collected at 1 min intervals for the
first 5 min after runoff generation, and then at 2 min and the sampling time was recorded with a stopwatch.
Surface runoff velocities were measured with KMnO$_4$ coloration. The 5 m long slope were divided into:
0-0.5 m, 0.5-1.5 m, 1.5-2.5 m, 2.5-3.5 m, 3.5-4.5 m, and 4.5-5 m. Among them, 0-0.5 m and 4.5-5 m are



used as transition areas. The average of the runoff velocity of the four sections was corrected (correction
factor 0.75) as the average flow velocity of the slope(Luk and Merz 1992). The water temperature was
measured with a thermometer. The runoff widths of the four sections were measured with a ruler. The
duration of each experiment was 45 min. The runoff and sediment samples were weighed, left for 24 h,
then the supernatant was poured off and transferred to aluminum boxes, dried in an oven at 105°C for 24
h and weighed to calculate the sediment amount. A digital camera (SONY A7RII) was used to take
photos of the slope surface before and after the experiment, and the overlap of each photo was required
to be at least 60%. Based on the 3D photogrammetry technique, high-precision DEMs data of the
experimental soil surface were obtained. After completing the slope photography, the experimental plots
were covered with plastic sheeting and left for 24 h until the next experiment.
**2.4 Data analysis**
**2.4.1 Rill hydrodynamic parameters**

Flow hydrodynamic parameters have a decisive role in the runoff and sediment production

characteristics of slope and are the basis for understanding the soil erosion processes and kinetic
mechanisms on slope(Cao et al., 2015). Therefore, commonly used hydrodynamic parameters such as
flow velocity ($V$), Reynolds number ($Re$), Froude number ($Fr$), Darcy-Weisbach coefficient ($f$), shear
stress ($\tau$), stream power ($\omega$) and unit stream power ($P$) are selected to evaluate the influence of slope($S$),
inflow rate($I$) and scouring times($N$) on rill erosion of spoil tips.

The Reynolds number ($Re$) and the Froude number ($Fr$) indicate the flow pattern and flow type of

the slope surface, and are calculated as follows(An et al., 2014):
$Re = \frac{Vh}{\gamma}$       (1)
$Fr = \frac{V}{\sqrt{gh}}$       (2)

where $V$ is the average flow velocity (m s$^{-1}$), $h$ is the flow depth (m), $h = \frac{q}{VbT}$, $q$ (m$^3$) is the total

amount of flow in a certain time $T$ (s), b is the width of surface flow (m), $\gamma$ is the kinematic viscosity (m$^2$
s$^{-1}$), $\gamma = \frac{0.01775}{1+0.0337t+0.000221t^2}$ , $t$ is the temperature of the water (°C) and $g$ is the gravitational acceleration
(9.8 m s$^{-2}$).



The Darcy-Weisbach coefficient (*f*) indicates the magnitude of resistance along the slope during
runoff flow and is calculated as follows(Abrahams et al., 1986):
$f = \frac{8gRJ}{V^2}$                                                                               (3)
where *f* is the Darcy-Weisbach coefficient, *J* is the hydraulic gradient (m m$^{-1}$), which can be
approximately replaced by the sine of the slope(Zhang et al., 2015), and *R* is the hydraulic radius (m),
which is often replaced by the flow depth.
Shear stress (*τ*) indicates the runoff scouring force that produces soil particle separation and
sediment transport and is calculated as follows(Nearing et al., 1991):
$\tau = \gamma_{\mathrm{m}}gR$                                                                        (4)
where *τ* is the shear stress (Pa),$\gamma_m$ is the mass density of the water–sediment mixture (kg m$^{-3}$).
The stream power (*ω*) indicates the power consumed by the flow acting on a unit area and is
calculated as follows(Govers et al., 2007):
$\omega = \tau V$                                                                                     (5)
where *ω* is the stream power (N m$^{-1}$ s$^{-1}$).
Unit stream power (*P*)is calculated as follows(Moore and Burch 1986):
$p = VJ$                                                                                              (6)
where $p$ is unit stream power (m s$^{-1}$).

**2.4.2 Rill morphology parameters**

The parameters of rill erosion such as rill depth, rill width and width-to-depth ratio were selected to
quantify the development characteristics of the rill network on the slope(Cerdan et al., 2002; Shen et al.,
2020) and to reflect the intensity of rill erosion along the vertical and horizontal directions. Based on the
3D photo reconstruction technology(Wu et al., 2018; Di Stefano et al., 2019), the photos taken by digital
cameras were imported into Agisoft Photoscan Professional 1.2.4 (Agisoft LLC, St. Petersburg, Russia)
for aligning photos, generating dense point clouds, generating grid textures, generating and exporting
DEMs, and so on. Afterwards, the DEMs (Resolution of 2 mm x 2 mm) data were imported into ArcGIS
software and the corresponding rill morphology parameters (e.g., rill width and rill depth) were obtained
with the help of mathematical and hydrological analysis functions in its spatial analysis.
The rill width and depth were extracted based on the 3D analysis method of ArcGIS, and a section
was selected at 0.5 m intervals starting from the top of the slope to extract the rill width and depth. The
average of the 10 sections was taken as the average width (*ARW*)and average depth (*ARD*). Based on the





hydrological analysis method of ArcGIS, a reasonable threshold of the cumulative amount of confluence
is set to initially extract the rills on the slope, and then compare the high-resolution photos taken in the
experiment to remove the non-existent fine rills.

The width-to-depth ratio of rill is an objective reflection of the variation in groove morphology(Tian

et al., 2020), and is calculated as follows:
$R_{WD} = \frac{ARW}{ARD}$ (7)

where $R_{WD}$ is the rill width–depth ratio, $ARW$ is the average width (cm), and $ARD$ is the average

depth (cm).

All data analysis was performed using the SPSS16.0 software (IBM Corp., Armonk, NY, USA).

Regression analysis was used to establish the equation simulation. Origin 8.5 software (Origin Lab Corp.,
Northampton, MA, USA) was used to visualize the data.
**3 Results**
**3.1 Runoff rate**

Fig. 4 illustrates the changes in runoff rate with time for three successive scouring at different slope

and inflow rates. According to Fig. 4, the runoff rate showed two characteristics variation: i.e., under the
lowest inflow rate (1.6 mm min$^{-1}$), the runoff rate increased with time in the early stages of the experiment
and then gradually stabilized. The runoff rate tends to increase and then fluctuate under relatively high
inflow rates (2 and 2.4 mm min$^{-1}$). At the early stages of the experiment, the low soil moisture content
and high soil infiltration rates result in low runoff rates. As the soil moisture content increased rapidly
and the soil infiltration rate decreased, runoff rates increased rapidly. When the soil infiltration rates
reach a stable stage, the runoff rates also became stable. Fluctuations in runoff rates are mainly related
to the development of rills (e.g., headward erosion, sidewall collapse and downcutting erosion) and are
relatively greater with increasing slope and inflow rate(Jiang et al., 2018). Overall, runoff rates increase
with slope, inflow rate and scouring times.

Regression analyses of slope, inflow rate and scouring times were performed to quantify their effects

on runoff rate. The best-fit equation to describe the mean runoff rate as a function of the adjusted slope,
inflow rate and scouring times is as follows:
$RR = 0.0024S^{1.1076}I^{1.8603}N^{0.3367}$     ($R^2$=0.9549, $P$<0.001, n=27) (8)





where *RR* is the mean runoff rate (mm min⁻¹), *S* is the slope (%), *I* is the inflow rate (mm min⁻¹) and
*N* is the scouring times.
The exponents in Eq. (8) are all positive, indicating that inflow rate, slope and scouring times all
have a positive effect on runoff rate. The exponents for slope, inflow rate and scouring times were 1.1076,
1.8603 and 0.3367, respectively. This indicates that the inflow rate plays an important role in the runoff
rate than the slope and scouring times.

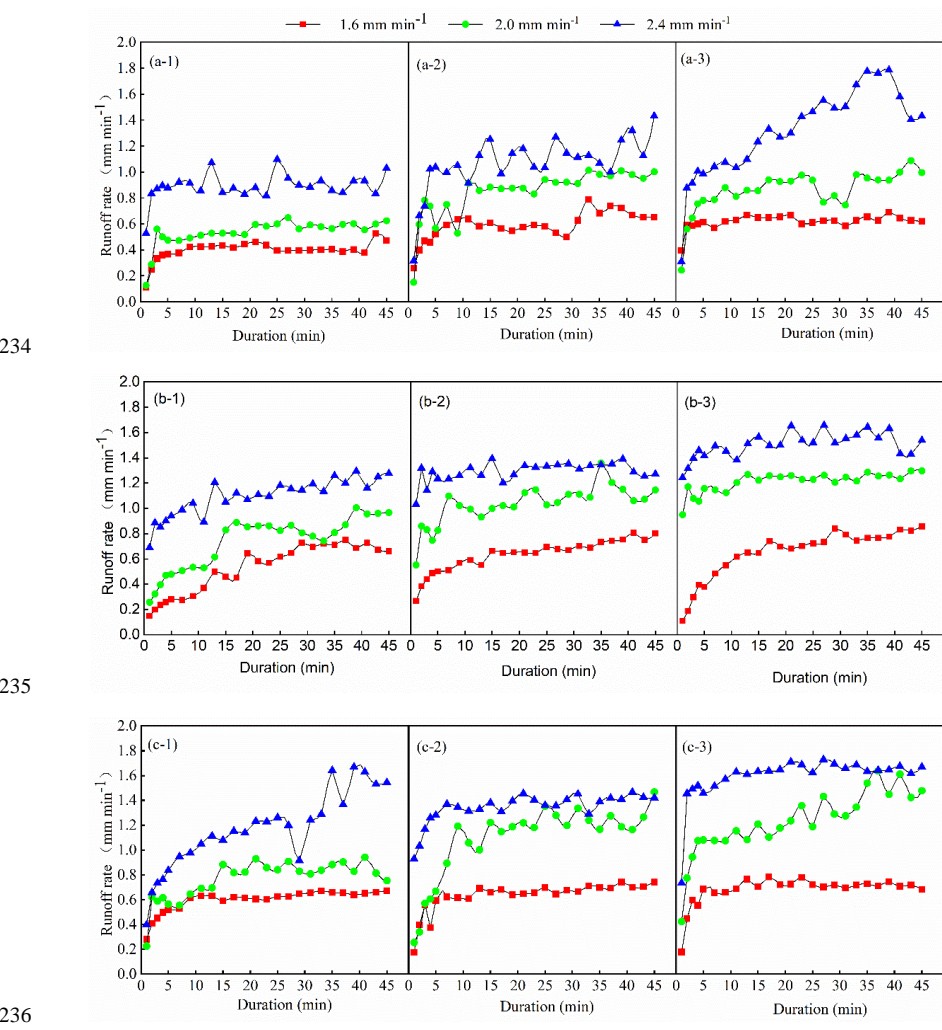




**Figure 4 Variations in the runoff rate with time for three scourings at slope of 28°(a-(1-3)), 32°(b-(1-3)) and**
**36°(c-(1-3)).**



### 3.2 Soil loss rate

The changes in three successive scouring soil loss rates with time for different inflow rates and slopes are shown in Fig. 5. Under the lowest inflow rate (1.6 mm min⁻¹), the soil loss rates were also low. The soil loss rates were large and fluctuated under relatively high inflow rates (2 and 2.4 mm min⁻¹). The higher the inflow rate and slope, the greater the fluctuation (Fig.5. a-1, b-1, c-1). In the process of slope erosion, the rill interconnection erosion intensified, and the side walls on both sides of the rill began to collapse (Fig.6). With the blocking and scouring of the side walls, the erosion and collapse occurred repeatedly, and erosion fluctuates, so that multiple peaks and lows occur during the erosion process(Peng et al., 2014; Niu et al., 2020). It is worth noting that for a given slope, the average soil loss rate increases with increasing number of scouring under the lowest inflow rate (1.6 mm min⁻¹). However, with the increase of inflow rate (2 and 2.4 mm min⁻¹), the average soil loss rate decreases with the increasing number of scouring. The reason may be due to the fact that at lower inflow rate (1.6 mm min⁻¹), runoff erosivity is relatively weak, and the rill networks gradually develop and mature with the number of scouring, resulting in an increase in average soil loss rates. However, at higher inflow rates (2 and 2.4 mm min⁻¹), the erosivity of runoff increases, and rill networks are basically mature after the first scouring, and as the number of scouring increases, the amount of material available for erosion decreases, leading to a decrease in average soil loss rates.

Regression analyses of slope, inflow rate and scouring times were performed to quantify their effects on soil loss rate. The best-fit equation to describe the mean soil loss rate as a function of the adjusted slope, inflow rate and scouring times is as follows:

$$SR = 0.0024 S^{1.6128} I^{2.8883} N^{-0.1777} \qquad (R^2{=}0.7955, P{<}0.001, n{=}27) \qquad (9)$$

where $SR$ is the mean soil loss rate (g m⁻² min⁻¹), $S$ is the slope (%), $I$ is the inflow rate (mm min⁻¹) and $N$ is the scouring times.

Eq. (9) shows that inflow rate and slope have a positive effect on soil loss rate, while the scouring times has a negative effect. The exponents for slope, inflow rate and scouring times were 1.6128, 2.8883 and -0.1777, respectively. This indicates that the inflow rate was the most important factor that affects soil loss rates.

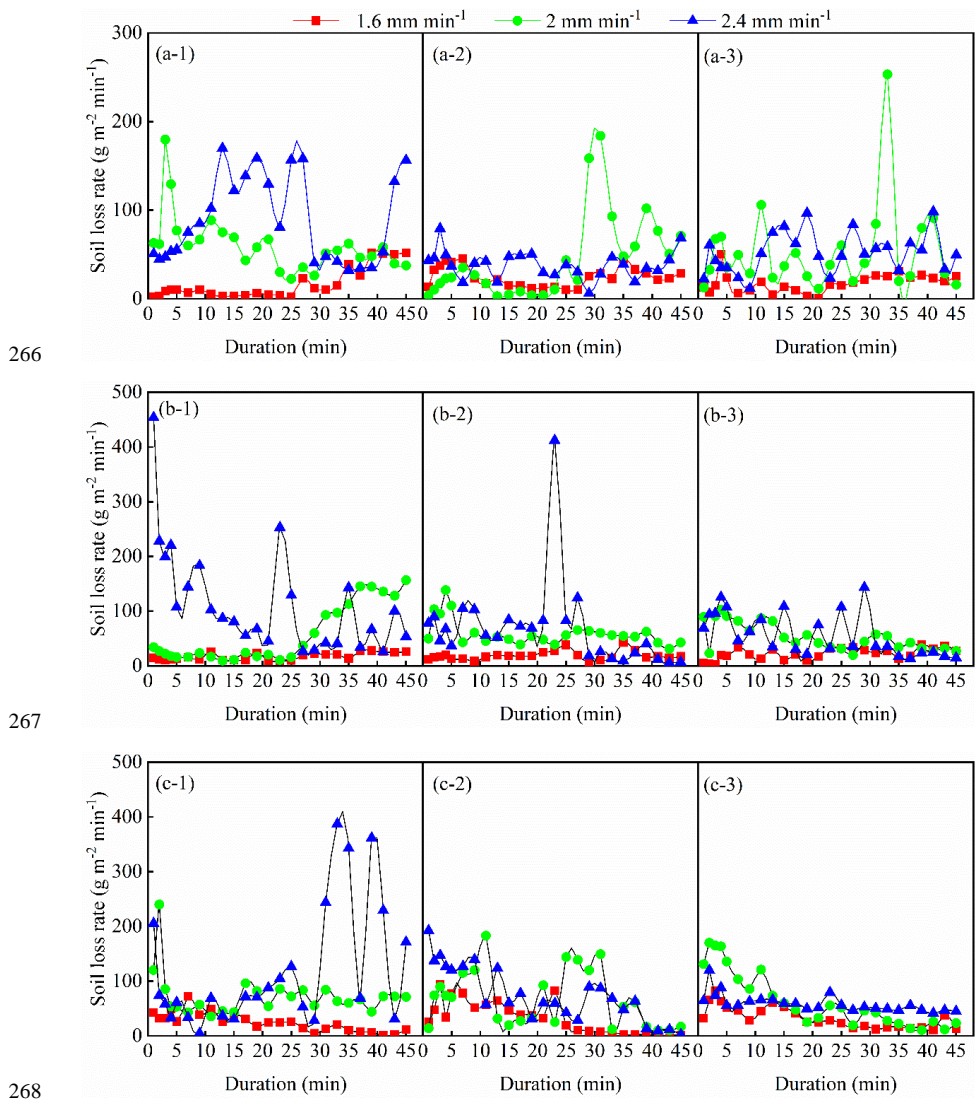




**Figure 5 Variations in the soil loss rate with time for three scourings at slope of 28°(a-(1-3)), 32°(b-(1-3)) and**
**36°(c-(1-3)).**





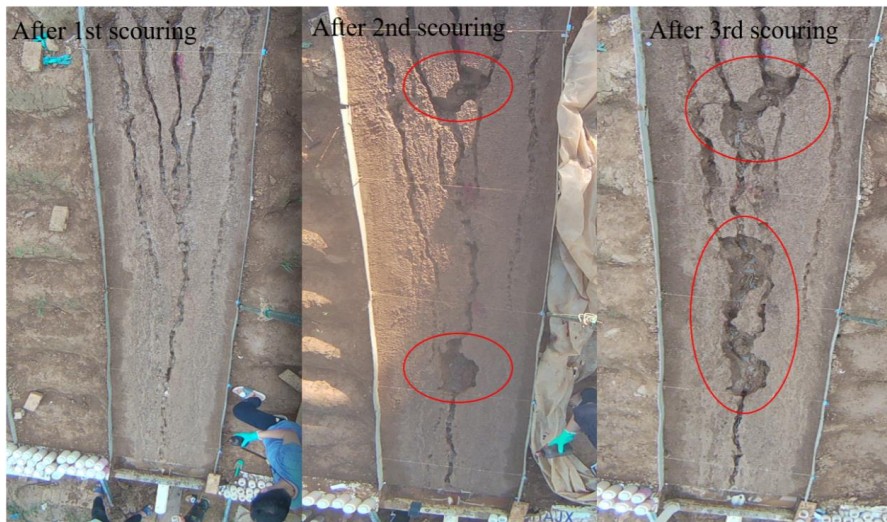


**Figure 6 Slope erosion after each scouring at the 2.4 mm min-1 inflow rate and a slope of 28 °.**
**3.3 Rill networks and morphology**
**3.3.1 Rill networks**
In order to explore the development of rill networks at different slopes, inflow rates and scouring
times, the rill network at the end of each experiment was shown in Fig.7. As can be seen from Fig.7,
there was significant variability in the development of rill networks at different slopes, inflow rates and
scouring times. When the inflow rate was low (1.6 mm min$^{-1}$), many intermittent rills and drop-offs
appeared on the slope as the scouring continues. As the number of scours and the slope increased, the
intermittent rills gradually becalmed connected along the slope to form continuous rills, and rill networks
becalmed relatively dense. In the process of the experiment, we observed that at the end of the third
experiment, the rills were still in the developmental stage, i.e., the rill network was not mature, which
may be related to the weak soil denudation capacity of the runoff (Fig.7 A-(1-3), D-(1-3), G-(1-3)). The
erosive force of the runoff increased with the inflow rate gradually (2 and 2.4 mm min$^{-1}$). Along with the
continuous scouring, the rill network on the slope has basically developed after the first experiment (Fig.7
B-1, C-1, E-1, F-1, H-1, I-1). In addition, the greater the inflow rate and slope, the faster the rill network
developed (Fig.7 I-1). We noted that at an inflow rate of 2 mm min$^{-1}$, the rill density was greatest (Fig.7
E-(1-3)), while at an inflow rate of 2.4 mm min$^{-1}$, the rill network was relatively sparse, suggesting that
there may be a critical inflow rate (2 mm min$^{-1}$) for the development of rills under the experiment





conditions. In general, the distribution of rills was denser at the top of the slope than at the bottom,
probably due to the main driver of soil erosion and rill development was upslope runoff with high erosive
capacity of the low sediment concentration(Tian et al., 2020). The overall predominance of parallel-
shaped rills at all experiments suggested that the formation of rills on slope were dominated by
concentrated runoff(Tian et al., 2017).

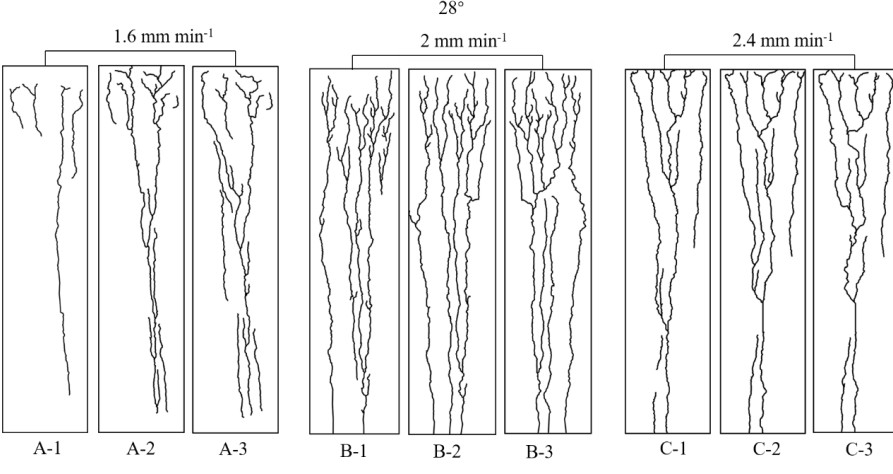


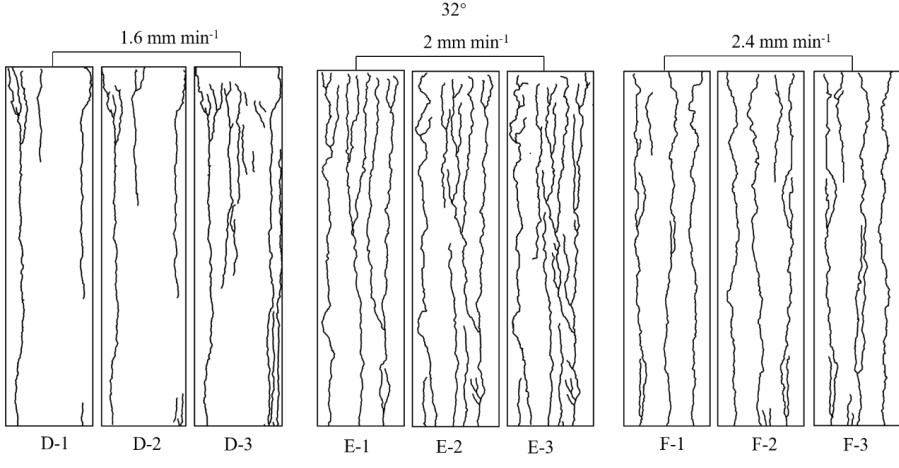


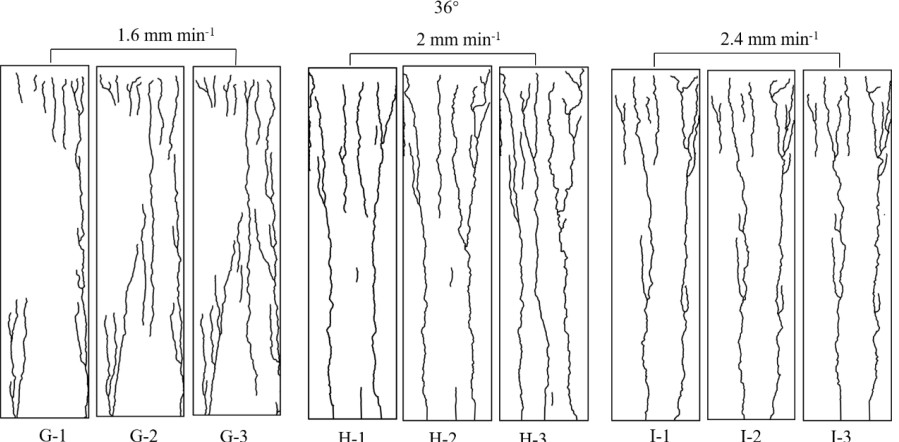


**Figure 7 Rill networks change at the end of each the experiment. A(1-3), B(1-3) and C(1-3) represent development of rill networks in three scouring, respectively under slope of 28°, inflow rates of 1.6, 2 and 2.4 mm min⁻¹. D(1-3), E(1-3) and F(1-3) represent development of rill networks in three scouring, respectively under slope of 32°, inflow rates of 1.6, 2 and 2.4 mm min⁻¹. G(1-3), H(1-3) and I(1-3) represent development of rill networks in three scouring, respectively under slope of 36°, inflow rates of 1.6, 2 and 2.4 mm min⁻¹.**

### 3.3.2 Rill characteristics

The mean rill width increased with the inflow rate and scouring times, however, decreased with increasing slope (Fig. 8(a-c)). The mean rill depth increased with the slope, inflow rate and scouring times (Fig. 8(d-f)). Furthermore, the increase in average rill depth was greater than the increase in average rill width. With the same inflow rate, the rill width-to-depth ratio decreased with increasing slope and scouring times (Fig. 8(g-i)), indicating that the increase in undercutting erosion of the rill significantly exceeds the collapse erosion of the rill wall, resulting in a decrease in the rill width-to-depth ratio.

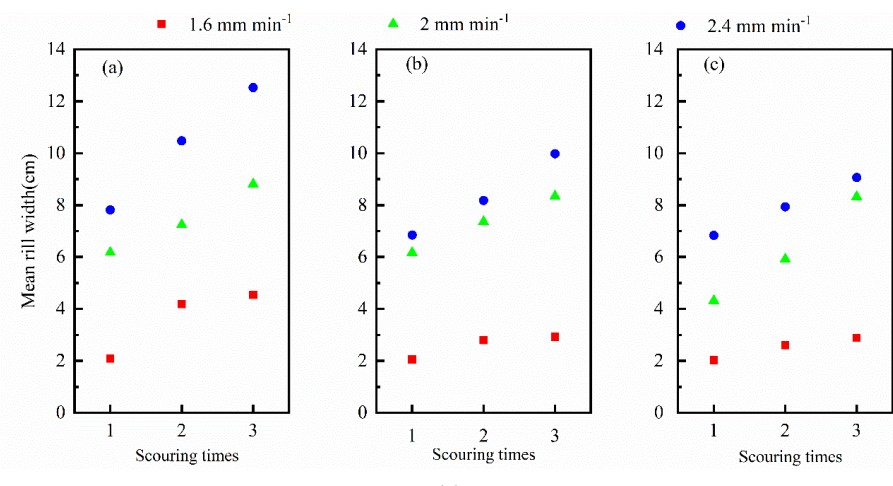



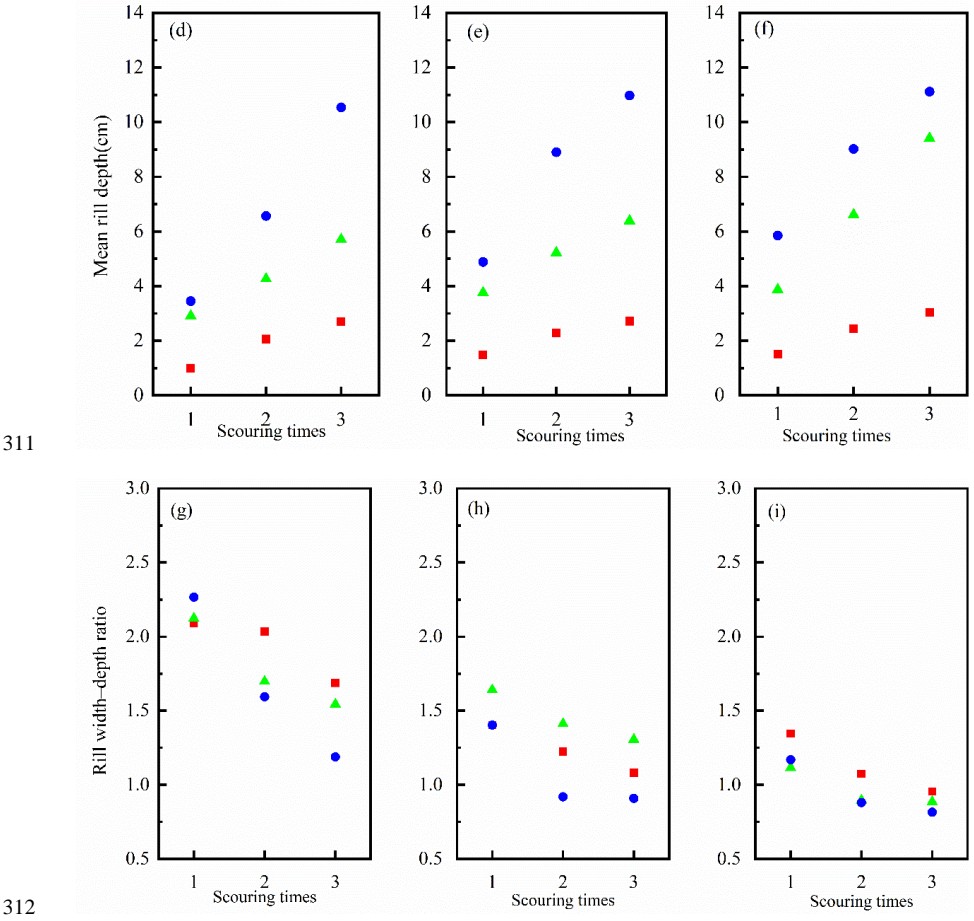

**Figure 8 Variation of rill characteristics with scouring times. Variation of the mean rill width with scouring times on slopes of 28°(a), 32° (b)and 36°(c). Variation of the mean rill depth with scouring times on slopes of 28°(d), 32° (e)and 36°(f). Variation of the rill width-depth ratio with scouring times on slopes of 28°(g), 32° (h)and 36°(i).**

Based on the above analysis, variation in rill morphological parameters were influenced by slope, inflow rate and scouring times, and regression analysis was used to quantify the effect of these influences on rill morphology. Eq. (10-12) shows that the average rill width, mean rill depth and rill width-to-depth ratio can be expressed as a power function of the slope, inflow rate and scouring times. Moreover, the fitted equations were all extremely significant (p<0.001). The coefficients indicate that the inflow rate has a greater effect on mean rill width and mean rill depth than slope and number of scouring, indicating that high inflow rate was to the main driver of the rill development(Niu et al., 2020). While scouring times has the greatest effect on the rill width-to-depth ratio.





$ARW = 59.1054S^{-1.471}I^{2.8454}N^{0.3979}$ ($R^2$=0.9147, $P$<0.001, n=27) (10)
$ARD = 0.0012S^{1.3988}I^{3.2357}N^{0.7070}$ ($R^2$=0.9607, $P$<0.001, n=27) (11)
$R_{WD} = 4.8324 \times 10^4 S^{-2.5427}I^{-0.3828}N^{-0.3079}$ ($R^2$=0.8911, $P$<0.001, n=27) (12)

328   where $S$ is the slope (%), $I$ is the inflow rate (mm min⁻¹), $N$ is the scouring times, $ARW$ is the mean

rill width (cm), $ARD$ is the mean rill depth (cm)and $RWD$ is the rill width–depth ratio.

330   The development of rill morphology is ultimately presented in terms of sediment. To reveal the

relationship between changes in rill morphology and sediment, data sets of rill morphological parameters
(mean rill width, mean rill depth and rill width–depth ratio) and cumulative sediment yield were analyzed
(Fig. 9). There is a quadratic function relationship between cumulative sediment yield and mean width
($R^2$=0.5337, $P$<0.01) (Fig. 9a) and width-depth ration ($R^2$=0.2327, $P$<0.05) (Fig. 9c). In addition, there
is a highly significant power function relationship between cumulative sediment yield and mean rill depth
width ($R^2$=0.5525, $P$<0.01) (Fig. 9b). In other words, the mean rill depth is the best indicator of rill
morphology to predict the production of sediment on slope.

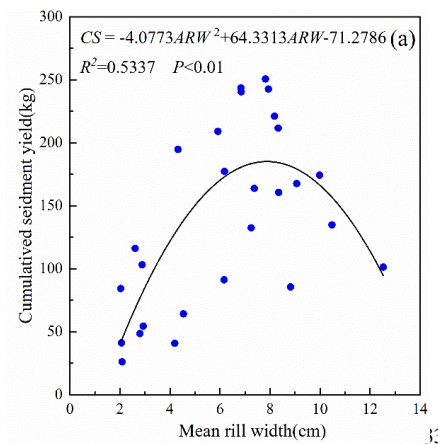

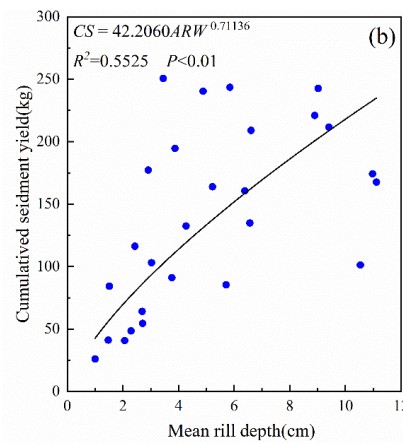

338                  339



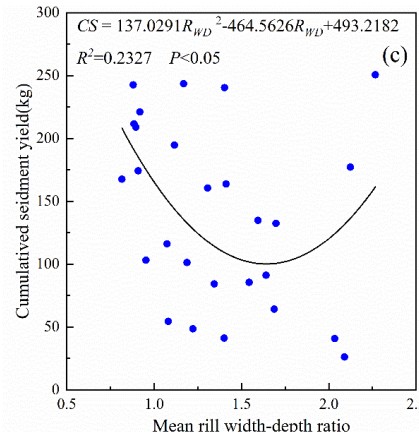

**Figure 9 Relationship between cumulative sediment yield and rill morphological parameters including mean**
**rill width (a), mean rill depth (b) and rill width-depth ratio (c).**
**3.4 Hydraulic characteristics and dynamic mechanisms of rill erosion**
**3.4.1 Rill flow hydraulic characteristics**
Rills formed are the result of concentrated runoff and that the analysis of the rill flow hydraulic
parameters can contribute to revealing the mechanism of rill erosion in spoil tips(Jiang et al., 2018). The
average rill flow velocity($V$) and Reynolds number ($R_e$) ranged from 0.18 to 0.30 m s$^{-1}$ and 178.85 to
1470.51 respectively, increasing with slope, inflow rate and scouring times (Fig. 10(a-c), (d-f)). The
Froude number ($Fr$) ranged from 1.16 to 1.93, all greater than 1(Fig. 10(g-i)). The reason for the lack of
a significant variable rule of $Fr$ with increasing slope, inflow rate and scouring times may be related to
the complexity of the rill morphological development on the slope. Based on the open-channel hydraulics
theory, flow regime could be classified into three types, namely laminar flow ($Re$<500), turbulent flow
($Re$>2000) and transitional flow (500<$Re$<2000). Moreover, $Fr$=1 distinguishes between subcritical and
supercritical flow. According to the results of Guo et al.(2020), the runoff under the experimental
conditions were of supercritical-laminar flow and supercritical- transition flow. The Darcy-Weisbach
coefficient ($f$) ranged from 1.14 to 3.15, no obvious relationship observed between $f$ and the slope, the
inflow rate and scouring times (Fig. 10(j-l)), the reason for which may be related to the rill beds becoming
more irregular, resulting in rill development(Jiang et al., 2018).





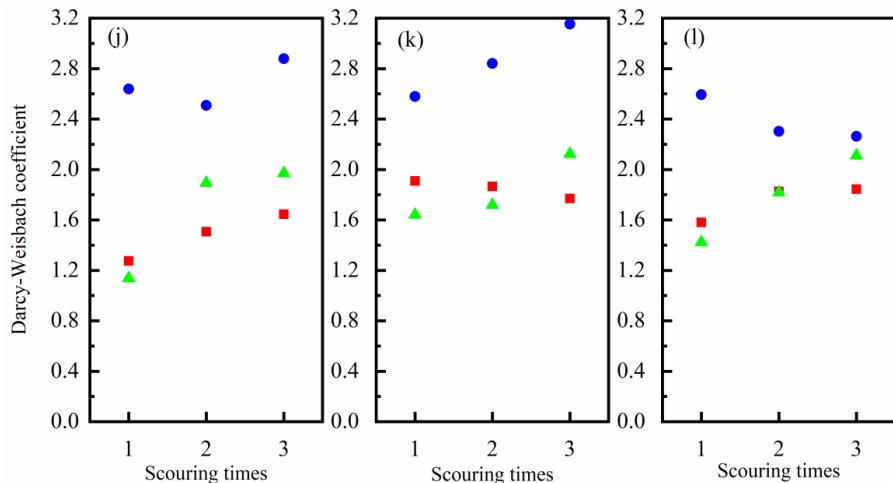

**Figure 10 Variation in rill flow hydraulic parameters. Variations in the mean flow velocity with inflow rate and scouring times at slope of 28°(a), 32°(b) and 36°(c). Variations in the Reynolds number with inflow rate and scouring times at slope of 28°(d), 32°(e) and 36°(f). Variations in the Froude number with inflow rate and scouring times at slope of 28°(g), 32°(h) and 36°(i). Variations in the Darcy-Weisbach coefficient with inflow rate and scouring times at slope of 28°(j), 32°(k) and 36°(l).**

To examine the relationship between changes in rill flow hydraulic characteristics and rill erosion, data sets of rill flow hydraulic parameters (mean flow velocity, Reynolds number, Froude number and Darcy-Weisbach coefficient) and soil detachment rate were analyzed (Fig. 11).The soil detachment rate ($D_r$) can be expressed as a power function of the velocity ($V$) ($R^2 = 0.4992$, $P < 0.01$) (Fig. 11a), Reynolds number ($Re$) ($R^2 = 0.6033$, $P < 0.01$) (Fig. 11b), Froude number ($Fr$) ($R^2 = 0.3969$, $P < 0.01$) (Fig. 11c) and Darcy-Weisbach coefficient ($f$) ($R^2 = 0.3981$, $P < 0.01$) (Fig. 13d), respectively. In other words, Reynolds number ($Re$) was the best hydraulic parameter to describe rill erosion on spoil tips.



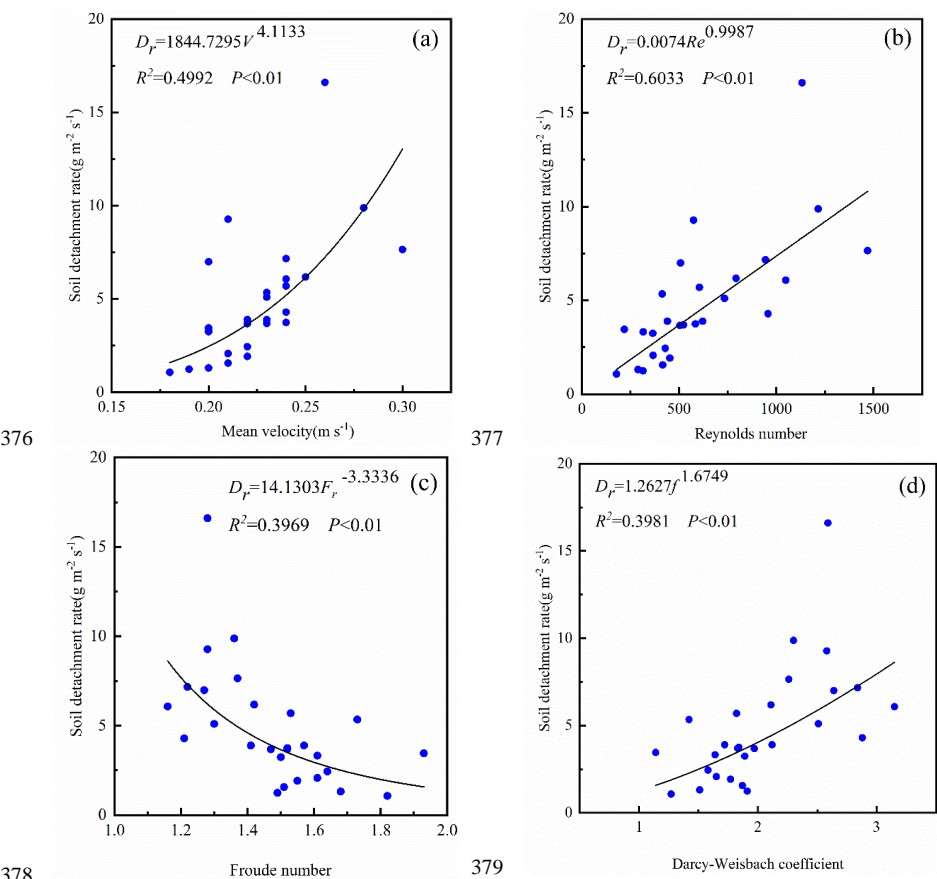

**Figure 11 The relationship between soil detachment rate and rill flow hydraulic parameters hydraulic parameters including flow velocity(a), Reynolds number(b), Froude number(c)and Darcy–Weisbach coefficient(d).**

**3.4.2 Hydrodynamic mechanisms of rill erosion**

The process of detachment and sediment transport by runoff is an energy-consuming. Therefore, in order to further reveal the mechanism of rill erosion, three hydrodynamic indicators were selected and calculated, as shown in Fig. 12. The mean shear stress ($\tau$), stream power ($\omega$) and unit stream power ($p$) ranged from 5.25 to 28.18 Pa (Fig. 12(a-c)), 0.95 to 8.45 N m$^{-1}$s$^{-1}$ (Fig. 12(d-f)), and 0.08 to 0.18 m s$^{-1}$ (Fig. 12(g-i)), respectively, and both of them increased with increasing slope, inflow rate and scouring times.





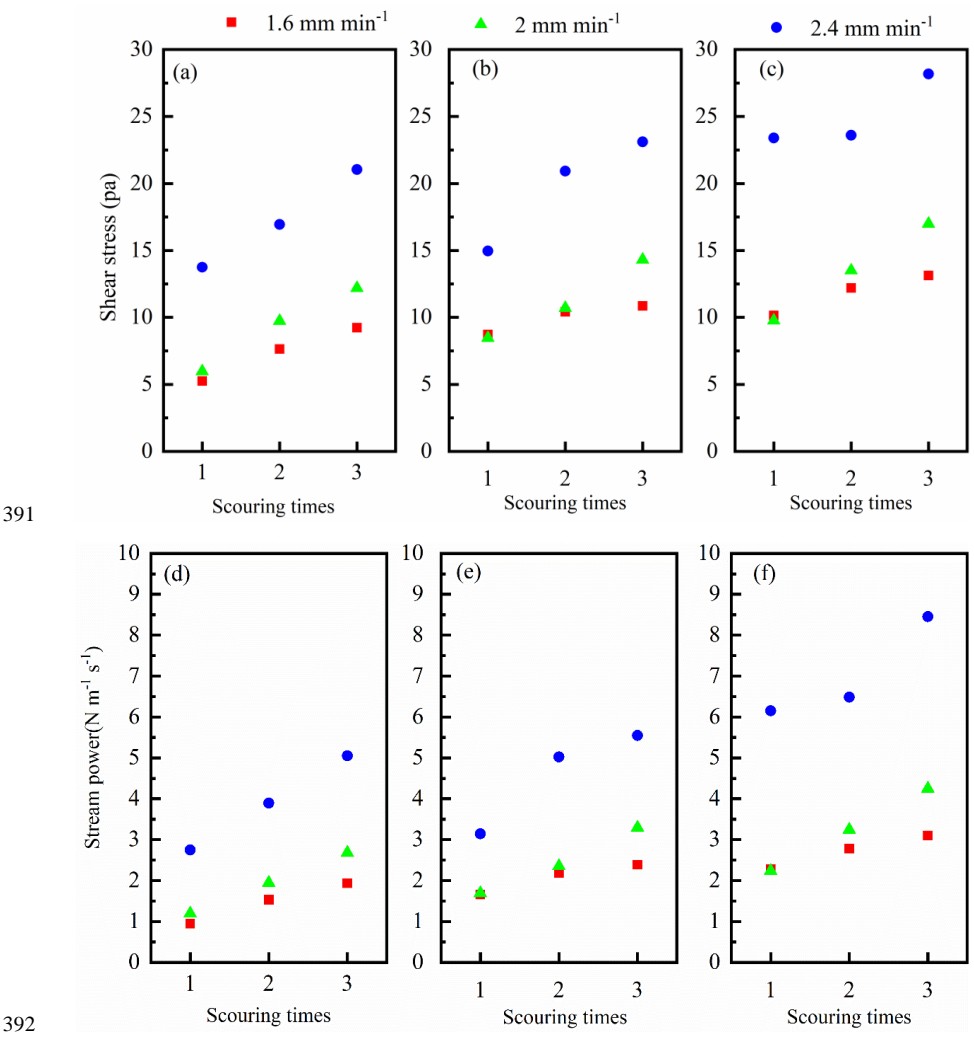

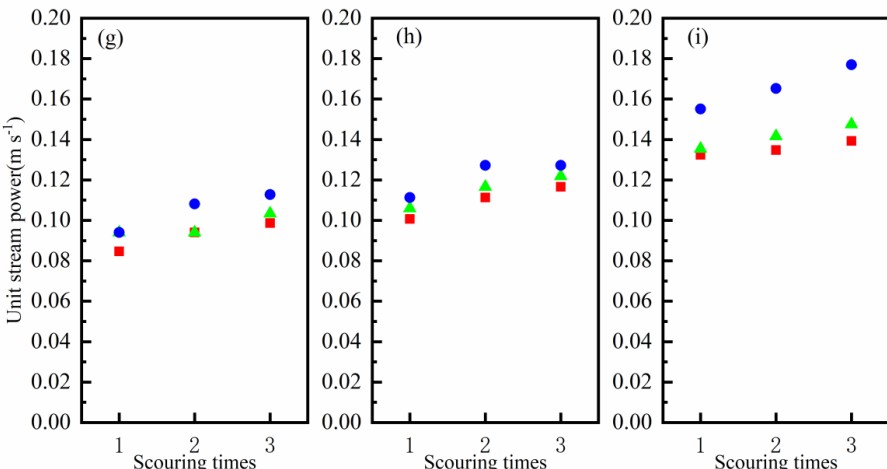

**Figure 12 Variation in rill flow hydrodynamic parameters. Variations in the shear stress with inflow rate and scouring times at slope of 28°(a), 32°(b) and 36°(c). Variations in the stream power with inflow rate and scouring times at slope of 28°(d), 32°(e) and 36°(f). Variations in the unit stream power with inflow rate and scouring times at slope of 28°(g), 32°(h) and 36°(i).**

To reveal the relationship between changes in rill flow hydrodynamic characteristics and rill erosion, data sets of rill flow hydrodynamic parameters (shear stress, stream power and unit stream power) and soil detachment rate were analyzed (Fig. 13).The soil detachment rate ($D_r$) can be expressed as a power function of the shear stress ($\tau$) ($R^2 = 0.6148$, $P < 0.01$) (Fig. 13a), stream power ($\omega$) ($R^2 = 0.6177$, $P < 0.01$) (Fig. 13b) and unit stream power ($p$) ($R^2 = 0.4075$, $P < 0.01$) (Fig. 13c), respectively. Furthermore, stream power ($\omega$) was the best hydrodynamic parameter to describe rill erosion on spoil tips.

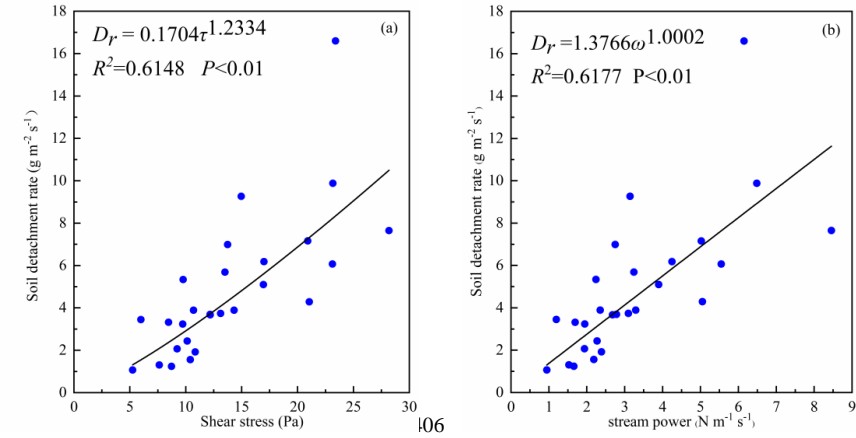

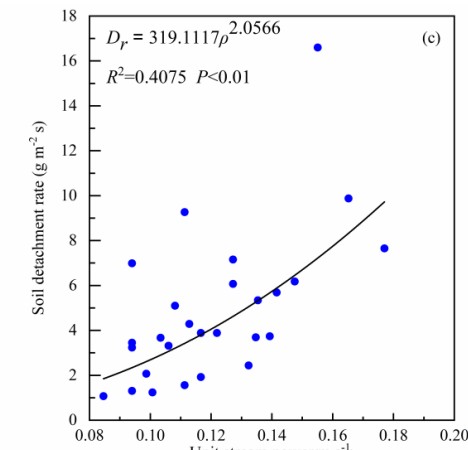

**Figure 13 Relationship between soil detachment rate (Dr) hydrodynamic parameter including Reynolds number (a), flow shear stress (b), stream power(c) and unit stream power(d).**

## 4. Discussion

### 4.1 Effects of slope, inflow rate and scouring times on runoff and soil loss

According to Eq. (8) and (9), the importance of slope ($S$), inflow rate($I$) and scouring times ($N$) on runoff rate and soil loss rate are in the following order: $I > S > N$. Inflow rate is the most important factor affecting soil erosion on the slope of spoil tips. The runoff coming from above the platform is involved in all aspects of soil erosion as a transmission link between the erosive power of the runoff and the energy of the flow on the slope, and can cause soil erosion on the steep slope(Zheng et al., 2000). The results of the study by Zheng et al. (2004) showed runoff and sediment from the upslope and rill flow hydraulic parameters have an important influence on rill sediment detachment and transport under the process of rill erosion. Therefore, in the management of soil erosion in spoil pits, the focus should be on how to effectively regulate runoff. For example, the use of vegetation measures to divide the spoil tips platform into a number of runoff dispersal units, so that heavy rainfall caused runoff is evenly dispersed among the units, this way it can effectively increase rainfall retention and infiltration, disperse runoff, dissipate runoff energy and reduce the soil erosion of the slope(Zhang et al., 2015; Zhang et al., 2016). For a given slope, vegetation or engineering measures(Pan and Ma 2020) can be used to regulate slope runoff and reduce its erosive energy to achieve soil and water conservation.

In addition, the effect of slope on runoff rate and soil loss rate is second only to the inflow rate. On the one hand, it is generally accepted that the greater the slope, the greater the partitioning of soil particles in the downhill direction, the less stable the soil particles and the more susceptible they are to erosion. On the other hand, an increase in the slope increases the runoff velocity(Tian et al., 2020) and reduces the residence time and infiltration time of the runoff on the slope, which increases the runoff and sediment yield on the slope. Wu et al. (2018) reported that sediment yield tends to increase with increasing slope. Therefore, it is necessary to take the slope factor into account when managing water and soil loss in spoil pits. For instance, slope grading and slope cutting to reduce slope lengths and slope,





together with vegetation and engineering measures, can reduce the probability of landslides and debris flows in heavy
rainfall conditions.

### 4.2 Rill networks and morphology characteristics

The development of rill on the slope mainly goes through a rill formation stage, a rill development stage and a

rill adjustment stage (Fig.14). The results of this study are similar to Jiang et al. (2018).After the first experiment, rill
network was basically formed, and the higher the inflow rate and the steeper the slope the more developed the rills
(Fig. 14 A-1,B-1). During the second experiment, with the initial formation of rills, the slope runoff mainly
converges to the outlet in the form of rill flow, during which the erosive force of the rill flow increases and the
headwater erosion of the downhills can proceed rapidly, forming a continuous rill (Fig. 14 C-2). Undercutting erosion
of the rill bottom and spreading erosion of the rill wall increase, and the rill depth and rill width increase (Fig. 14 A-
2,B-2). In the third experiment, the rill flow adjusted some parts of the already developed rill. The bottom and inner
walls of rills were mainly scoured, and the rills collapsed due to the hollowing of the walls by the rill flow, which
caused rills to become less stable under gravity (Fig. 14 A-3,B-3, C-3).

The overall predominance of parallel-shaped rills at all three experiments (Fig. 7) is consistent with the findings

of Fang et al. (2015) and Tian et al. (2020). However, Shen et al.(2020) showed that the rill network mainly exhibits
a dendritic pattern. The difference may be due to the fact that slope surface flow under scour conditions is surface-
produced flow and is point-produced flow under rainfall conditions(Zhang et al., 2013), and the difference in the way
they produce flow may lead to a different development of the rill network. The high clay content (31.15%) of the soils
in this experiment results in relatively strong inter-soil adhesion and resistance to erosion by runoff, but the poor
infiltration rate results in relatively high runoff volumes, and the slopes often form multiple rills of approximately
parallel width and depth.

The most eroded parts of the slope under scour conditions are mostly located in the middle and upper parts of the

slope (Fig. 7,14). This result is similar to that obtained by Yang et al. (2019), who noted that the highest proportion of
rill erosion was generated on the upper part of the slope, reaching over 60 %. The reason is that the flow and erosion
forces are greatest when the water enters the slope from the top of the slope, thus the rill head appears at the top of the
slope, and once the drop can begin to appear and develop into a rill at the top of the slope, rill erosion will rapidly
undergo headwater erosion, undercutting erosion and lateral erosion. Sediment yield, rill width and depth increase
rapidly. As the runoff infiltrates and is subjected to resistance along its course, the energy of the runoff is gradually
depleted and the increased sediment content of the runoff reduces its separation capacity, which in turn reduces the
proportion of rill erosion in the lower part of the slope. However, under rainfall conditions the most severe erosion is
observed in the middle and lower parts of the slope(Jiang et al., 2018). The reason for this is that under rainfall
conditions, the runoff gradually tends to increase along the slope length, the runoff velocity increases, and the ability
of the runoff to strip the soil increases as well. The lower and middle parts of the slope are prone to the development
of rills.

The rill depth and cumulative sediment yield exhibited significant power relationship (Fig. 9b). The mean rill

depth is the best rill morphological parameter for predicting sediment. However, the results of  Niu et al. (2020)





showed that cumulative sediment yield can be expressed as a power function of cross-sectional area. Shen et al. (2015)
investigated the development of rill networks and the quantitative description of rill morphology through continuous
rainfall experiments. The results showed that the mean rill width was the best basic morphological indicator for
evaluating rill erosion. Differences in experimental methods, soil types, rainfall conditions and topography may have
contributed to the above differences in the results.

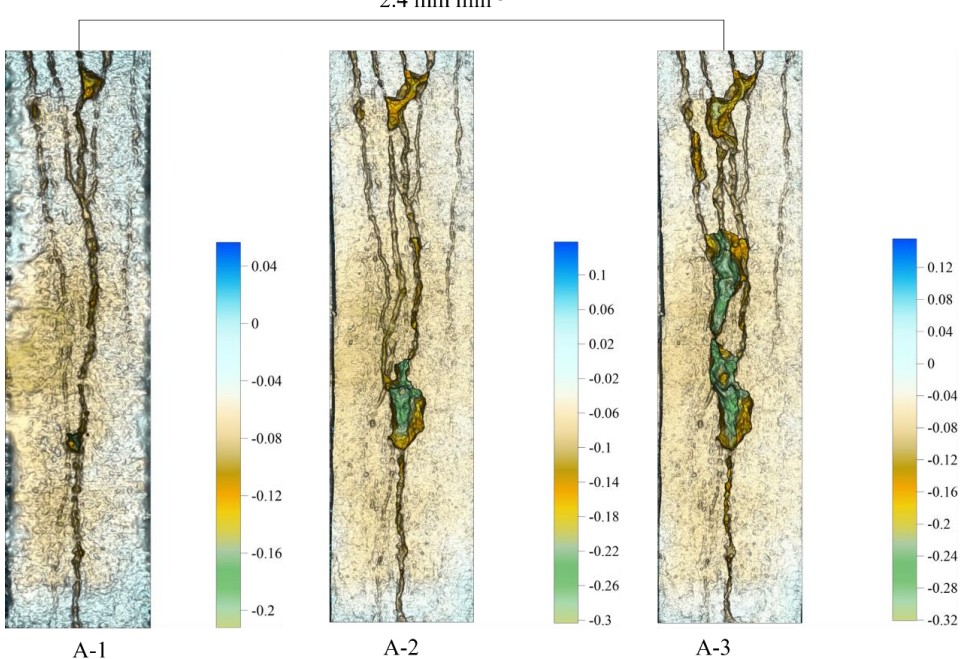




32°

2.4 mm min$^{-1}$

B-1          B-2          B-3


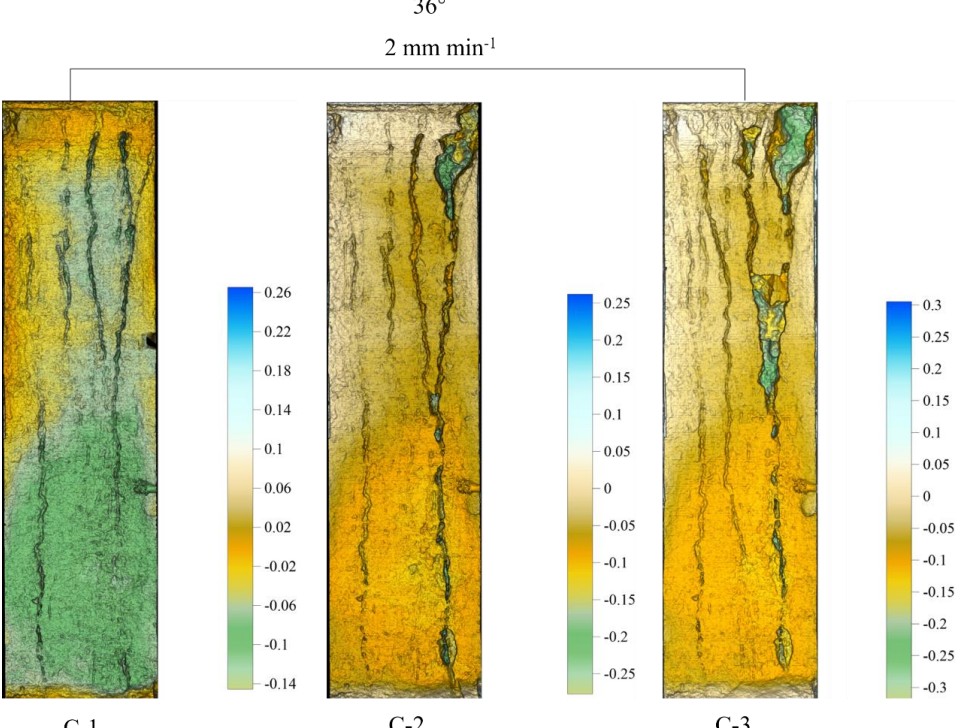


**Figure 14 DEMs of different slope, inflow rate and scouring times. A(1-3) represent change of DEMs in three scouring, under slope of 28°, inflow rates of 2.4 mm min-1. B(1-3) represent change of DEMs in three scouring, under slope of 32°, inflow rates of 2.4 mm min-1. C(1-3) represent change of DEMs in three scouring, under slope of 36°, inflow rates of 2 mm min-1.**

**4.3 Hydraulic characteristics and dynamic mechanisms of rill erosion**
The runoff hydrodynamic characteristics largely determine the rill erosional and morphological characteristics
on slopes. The runoff hydrodynamic characteristics can describe the energy changes in runoff (Xiao et al., 2009),
which in turn have an impact on the stripping, transport and deposition of soil on slopes. The Reynolds number is the
best hydraulic parameter for predicting rill erosion (Fig. 11b). This result is similar to that obtained by Guo et al.
(2018),who found the Reynolds number ($Re$) was the best predictor for sediment load. However, (An et al.,
2014)considered the Froude number($F_r$) as a key hydraulic parameter affecting soil loss, because the Froude number
($F_r$) is the ratio of inertial forces to gravitational forces, and these forces were closely related to sediment concentration.
Li et al. (2016), Shen et al. (2016) and Jiang et al.(2018) considered that among the various hydraulic parameters, the
flow velocity ($V$) best represents the hydraulic characteristics of the rill flow. The process of runoff stripping and
transporting of soil is actually a process of doing work and consuming energy. Therefore, in the process of rill
development, changes in hydrodynamic characteristics play an important role in the erosion characteristics of rill
runoff. Our results show that stream power ($\omega$) was the best hydrodynamic parameter to describe rill erosion
mechanism (Fig. 13b), which is consistent with the results of Al-Hamdan et al. ( 2012) and Niu et al. (2020). But, Li



et al. (2016) considered that shear stress provides the best characterization of hydrodynamic parameters in rill erosion.
**5 Conclusions**
The rill erosion process, the rill morphological characteristics and the rill erosion hydrodynamic mechanism of
spoil tips, were studied by multiple scouring experiments in the field. The results showed that the importance of
slope($S$), inflow rate($I$) and scouring times($N$) on runoff rate and soil loss rate are in the following order: $I > S > N$,
indicating that inflow rate was the most important factor affecting rill erosion on the slope of the spoil heaps. Therefore,
in the management of soil erosion in spoil tips, the focus should be on how to effectively regulate runoff from the
platform and slope.
The development of rill mainly goes through three stages: the rill formation stage, the rill development stage and
the rill adjustment stage. The overall predominance of parallel-shaped rills at all experiments suggested that the
formation of rills was dominated by concentrated runoff. The most eroded parts of the slope were mostly located in
the middle and upper parts of the slope of spoil tips. Rill depth was the best rill morphological parameter for evaluating
spoil tips rill erosion.
The Reynolds number ($Re$) and stream power ($\omega$) were the best hydraulic parameter and hydrodynamic parameter
for predicting rill erosion, respectively. The study has some importance practical implications for the management of
soil erosion and the establishment of erosion prediction models for spoil tips.

**Acknowledgments.** This work was supported by the National Natural Science Foundation of China (Grant No.
41671283 and 2016YFC0501706-02).

**Author contribution**. Zhaoliang Gao and Yongcai Lou designed the experiment. Fuyu Zhou, Jianwei Ai, Yunfeng
Cen, Tong Wu and Jianbin Xie carried out the experiment. Yongcai Lou prepared the manuscript with contributions
from all co-authors.
**Data availability.** Not applicable.
**Compliance with ethical standards**
**Competing interests.** The authors declare that they  have no conflict of interest.
**Ethical approve.** Not applicable.
**Consent to publish.** The authors confirm that final version of the manuscript has been reviewed, approve and
consented for publication by all authors.

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
