# Peer review of "Rill Erosion on Slope of Spoil tips: experimental study of runoff scouring erosion in multiple times"

_Hydrology and Earth System Sciences, 2021_

## Author Comment (AC7)

**Dear Reviewer:**

Thank you for taking the time to review our manuscript entitled "Rill Erosion on Slope of Spoil tips: experimental study of runoff scouring erosion in multiple times" (ID: hess-2021-399), and provide constructive comments. These comments are valuable and very helpful for revising and improving our paper, as well as the important guiding significance to our further researches. We have studied and analyzed comments carefully and have made many corrections which we hope meet with approval. To clearly respond all comments, point by point, these comments from reviewers were classified by authors based on the specific meanings and listed as following (Q1, Q2, Q3…). The main revisions in the paper and the responds to the reviewer's comments are as following.

**Reviewer #RC3:**

Hydrology and Earth System Sciences

Manuscript Number: hess-2021-399

Article Type: Research paper

**Q1.** it is not clear why only three slopes were considered and why those and not others.

**Response:** we are so sorry for our unclear expression. In fact, we have investigated the material composition, slope and slope length of 368 spoil tips in different regions of China. The production and construction projects which were selected for the survey are all highly representative in the regions where they are located, and the results of the survey can reflect the basic conditions of the regional spoil tips in more comprehensively. The survey results show that the slope range of the spoil tips is generally between 25-40°, with 28-36° accounting for more than 75 % of the total survey. Therefore, we have gradients to set three slopes of 28°, 32° and 36° respectively. In any field experiments of this nature, one has to select some typical slopes that best represent the real-world situation and in our case the choice of the three slopes was made based on survey of real-world spoil tips.

**Q2.** it is not clear why only three types of inflow rates (surprisingly not rainfall) are considered.

**Response:** Spoil tips are a unique man-made mound landform formed by production and construction activities, with a "platform-steep slope" structure. Under field conditions, the runoff collected by the compaction platform of spoil tips is an important factor causing slope scour erosion and accelerating erosion of engineered landscapes. The infiltration rate of the platform formed by heavy mechanical rolling is significantly reduced. Under rainfall conditions, the platform produces large concentrated runoff preferentially over the slope, and concentrated runoff rapidly flows along the edge of the platform to the steep slope, thus causing severe slope erosion. Therefore, the runoff collected by platform of spoil tips is an important factor causing slope scour erosion and accelerating erosion of engineered landscapes (Zhang et al. 2015, Zhang et al. 2016). The amount of runoff from the platform is determined by the rainfall intensity. Three rainfall intensities of 1.6, 2, and 2.4 mm min$^{-1}$ were used to mimic typical erosive rainfall from heavy, torrential and extremely heavy rain in study area, respectively. Rainfall intensities of 1.6, 2 and 2.4 mm min$^{-1}$ were converted to equivalent inflow rate of 8, 10 and 12 L min$^{-1}$ according to the amount of water, respectively.

- Zhang LT, Gao ZL, Yang SW, Li YH, Tian HW (2015) Dynamic processes of soil erosion by runoff on engineered landforms derived from expressway construction: A case study of typical steep spoil heap. Catena 128, 108-121. https://10.1016/j.catena.2015.01.020.

- Zhang LT, Gao ZL, Li ZB, Tian HW (2016) Downslope runoff and erosion response of typical engineered landform to variable inflow rate patterns from upslope. Nat Hazards 80, 775-796. https://10.1007/s11069-015-1996-z.

**Q3.** the soil is clay loam; therefore, the entire work is soil type-specific (as however all the papers published on a similar approach).

**Response:** Clay loamy soil is a typical soil texture in study area (Guanzhong Plain of Loess Plateau) (Wu et al. 2021). Therefore, clay loam be used to simulate soil erosion of the spoil tips, and the research results are representative of the soil properties in the area.

- Wu J, Qi Y B, Chang Q R, Liu M Y, Bai L M (2021) Attribution of Lou Soil in

Chinese Soil Taxonomy and Establishment of Representative Soil Series in Guanzhong Area. Acta Pedologica Sinica, 58 (2):357-371. https://10.11766/trxb201906240325.

**Q4.** The real natural conditions where also vegetation play a role in soil erosion due to the roots are not considered, therefore the work is affected at its basis by a lack of representativeness of real conditions, and overall is affected by an "anthropogenic" setting of soil into a given plot. Differently, the approaches with natural soil, where also vegetation is present, with natural rainfall scenarios and more or less natural slope, are more representative of reality.

**Response:** We are in complete agreement with you. Natural slopes often have vegetation growing on the surfaces because there are no human influences. Under natural rainfall conditions, vegetation can effectively prevent the occurrence of soil erosion on slopes due to vegetation roots. However, spoil tips are a unique geomorphological unit which is formed by the artificial accumulation and remodeling of soil generated during the production and construction activities, which results in non-vegetation growth on the slope of the spoil tips at the early stage of formation, and it takes some time for the vegetation to recover naturally. In addition, the construction time of production and construction projects always rain heat synchronization, which lead to this stage that is the most serious stage of soil erosion in the spoil tips (no or little vegetation cover). In addition, serious erosion rates induce sediment and nutrients losses on discarded soils which make the disturbed soil accumulation vegetation reconstruction very difficult (Cerdà and García-Fayos2002). Therefore, soil erosion was often dominated by bare slope of the spoil tips, we mainly study and focus on the soil erosion of the spoil tips without vegetation cover stage. In addition, we thank for your suggestion and will focus on the effect of vegetation in the subsequent study to manage the erosion of the slope of the spoil tips.

The way in which spoil tips is formed causes it to differ from the natural slope or soil. Spoil tips has unique underlying surface conditions, for instance, destruction of soil structure, the vegetation degeneration, soil organic matter and plant root system deficiency, which result in its poor scouring resistance(Zhang et al.2015). The two

different subsurface conditions lead to differences in erosion. In addition, the driving force of soil erosion on natural slopes is mainly slope runoff formed by rainfall, while runoff from platforms is the main driving force of slope erosion on spoil tips. Therefore, we used runoff plots to study soil erosion on the slopes of spoil tips using artificial soil mounding and runoff scouring.

- Cerdà A, GarcíA-Fayos P (2002) The influence of seed size and shape on their removal by water erosion. Catena 48(4):293–301. https://doi.org/ 10.1 016/S0341-8162(02)00027-9.

- Zhang, L.T., Gao, Z.L., Yang, S.W., Li, Y.H., Tian, H.W(2015). Dynamic processes of soil erosion by runoff on engineered landforms derived from expressway construction: a case study of typical steep spoil heap. Catena 128, 108–121.https://doi.org/10.1016/j.catena.2015.01.020.

**Q5.** However, the work in its present form doesn't meet the high standard required for HESS, where too limited studies are not welcomed. Second, the work is too narrow and site-specific in its purpose, a fact that is given at the eyes of the readers an idea of a not representative analysis, therefore with findings impossible to generalize.

**Response:** Thank you for your valuable comments. These have not only greatly improved our work, but also learned more knowledge of experimental research and future research directions that need attention from your comments. We also believe that the article can be revised according to your comments to reach the hess publication level.

Soil erosion is a global environmental problem that has greatly hindered region's sustainable socioeconomic development. Especially nowadays, artificial soil erosion often results in severe environmental and economic problems such as degradation of agricultural soil and surface water quality, and damage to infrastructure and transportation corridors. Spoil tips are now the most significant type of artificial soil erosion in the world. The erosion of spoil tips has caused a large number of environmental problems. On the one hand, spoil tips are new landforms created by man-made action, which have destroyed the original surface structure (Zhang et al.2015; Migońand Latocha 2018), and caused geological disasters such as landslides

and mudslides (Iqbal et al.2018; Jiang et al.2018). These disasters thus seriously threaten the safety of human life (Nearing et al.2017; Conforti and Ietto 2020). For example, a large landslide accidentally occurred in Shenzhen City (China) in 2018, resulting in a death toll of 73(Gao et al. 2019). On the other hand, soil and water loss from spoil tips leads to substantial sediment yields transported into rivers, affecting river flood safety and water quality. Therefore, there is an urgent need to investigate the soil erosion for spoil tips because they cause serious damage and threaten human lives. Therefore, this study has a strong international dimension and has many readers. In addition, our study focuses on the erosion processes and mechanisms of spoil tips, which is consistent with the scope reported by hess.

- Zhang LT, Gao ZL, Yang SW, Li YH, Tian HW (2015) Dynamic processes of soil erosion by runoff on engineered landforms derived from expressway construction: a case study of typical steep spoil heap. Catena 128:108–121.https://doi.org/10.1016/j.catena.2015.01.020.

- MigońP, Latocha A (2018) Human impact and geomorphic change through time in the Sudetes, Central Europe. Quat Int 470:194–206.https://doi.org/10.1016/j.quaint.2018.01.038.

- Iqbal J, Dai FC, Hong M, Tu XB, Xie QZ (2018) Failure mechanism and stability analysis of an active landslide in the Xiangjiaba Resevoir area, Southwest China. J Earth Sci 29(3):646–661.https://doi.org/10.1007/s12583-017-0753-5.

- Jiang YH, Lin LJ, Chen LD, Ni HY (2018) An overview Nearing of the resources and environment conditions and major geological problems in the Yangtze River economic zone, China. China Geol 1(3):435–449.https://doi.org/10.31035/cg2018040

- Nearing MA, Xie Y, Liu BY, Ye Y (2017) Natural and anthropogenic rates of soil erosion. Int Soil Water Conserv 2:77–84. https://doi.org/10.1016/j.iswcr.2017.04.001.

- Conforti M, Ietto F (2020) Influence of tectonics and morphometric features on the landslide distribution: a case study from the Mesima Basin (Calabria, South Italy). J Earth Sci 31(2):393–409.https://doi.org/10.1007/s12583-019-1231-z.

- Gao Y, Yin YP, Li B, He K, Wang XL (2019) Post-failure behavior analysis of the Shenzhen"12.20" CDW landfill landslide. Waste Manag 83:171–183https://doi.org/10.1016/j.wasman.2018.11.015.

**Q6.** On the other hand, in the case of an established plot on natural soil (even covered by vegetation) respecting the real geomorphologic conditions (usually for these sites, few non-invasive fences and one outled/tank collecting water/sediment are enough to guarantee the experiment), the analysis is conducted with real rainfall conditions (not with forced inflow rate), even for one year.

**Response:** We are in complete agreement with you. The construction of runoff plots on natural soil by using natural rainfall observation is one of the methods to study soil erosion of spoil tips, but there are many disadvantages: first of all, spoil tips are usually located in sparsely populated ravines, and the roads are often damaged (because of flash floods and landslides) after rainfall, especially heavy rainfall, so that people are not able to arrive in the first place, which affects the accurate acquisition of test monitoring data. Second, the lack of electricity and signals makes it impossible to provide energy and data transmission for monitoring equipment. Third, in heavy rainfall conditions, field site monitoring may pose a threat to personal safety. Comprehensive consideration, we did not use the method of constructing runoff plots on natural slopes to study soil erosion of spoil tips. Based on the field investigation (slope, slope length, soil bulk density and soil type), the designed runoff plots are able to more realistically approximate the natural conditions in the field. It is feasible to study the rill erosion on the slope of the spoil tips by designing runoff plots.

Under field conditions, the runoff collected by the compaction platform of spoil tips is an important factor causing slope scour erosion and accelerating erosion of engineered landscapes (Zhang et al. 2015, Zhang et al. 2016). Runoff from the platform has a greater impact on soil erosion than rainfall in the spoil tips. Therefore, we used runoff scouring to study the effect of runoff from the platform on the erosion of the spoil tips.

- Zhang LT, Gao ZL, Yang SW, Li YH, Tian HW (2015) Dynamic processes of soil erosion by runoff on engineered landforms derived from expressway construction:

A case study of typical steep spoil heap. Catena 128, 108-121. https://10.1016/j.catena.2015.01.020.

- Zhang LT, Gao ZL, Li ZB, Tian HW (2016) Downslope runoff and erosion response of typical engineered landform to variable inflow rate patterns from upslope. Nat Hazards 80, 775-796. https://10.1007/s11069-015-1996-z.

**Thanks-note:** We really thank you for providing so many review comments. We have a deeper understanding of how to make the simulation experiment research fit the actual situation and apply the research results to reality. We thank you very much. Due to our limited level of research, if our answer is not good enough, and our responses makes you dissatisfied, we hope to get further suggestions. Thank you very much!

---

## Author Comment (AC8)

Dear Reviewer:

Thank you for taking the time to review our manuscript entitled "Rill Erosion on Slope of Spoil tips: experimental study of runoff scouring erosion in multiple times" (ID: hess-2021-399), and provide constructive comments. These comments are valuable and very helpful for revising and improving our paper, as well as the important guiding significance to our further researches. At the same time, we thank you for giving us the opportunity to revise the manuscript. We have studied and analyzed comments carefully and have made many changes which we hope adequatelly address the concerns of the reviewer. To clearly respond to all comments, point by point, these comments from reviewer were classified by authors based on the specific meanings and listed as following (Q1, Q2, Q3…). The main revisions in the paper and the responds to the reviewer's and editor's comments are as following.

**Reviewer #RC3:**

Hydrology and Earth System Sciences

Manuscript Number: hess-2021-399

Article Type: Research paper

**Q1.** A really very good piece of experimental work at a scale sufficiently large to provide insights into field scale processes. Well done!

**Response:** We first thank you for your recognition of our study and your positive decision. Also, the following comments are valuable for improving our paper. Thank you again.

**Q2.** However, the paper is quite difficult to follow in many places and the English needs a thorough rework, right from the title and onwards.

**Response:** We appreciate your suggestions very much. Based on the content of this paper, we have revised the title and changed the original title to "A runoff scouring experimental study of rill erosion of spoil tips". In addition, we have also improved the English.

Comments:

**Q3.** -The introduction sets the scene well. However, the paper neglects the fantastic work done by RS Parker and the work of Schumm, Mosely and Weaver (Experimental

Fluvial Geomorphology). It is imperative that this work be examined and referenced in this paper as there are many similarities.

**Response:** We are very grateful to RS Parker, Schumm, Mosely and Weaver the excellent contribution to drainage basin evolution and fluvial geomorphology. We have added references in this article.

**Q4.** - Also, there is I believe considerable debate regarding rill and rill measurement and characteristics. Given that you used state of the art survey methods, why didn't you extract cross-sections and compare your data with field measured cross-section from material in your local area or from other published data (i.e. from the references above and other data)?

**Response:** We are very grateful for your suggestion. We have added the data about rill cross-sections (Section 3.3.3) and a comparison with existing research data in the discussion. Thank you again.

**Q5.** -It was not clear about the rationale and timing of the 3 experiments for each slope. How can you ensure that antecedent soil moisture is the same for all? This is not a show-stopper in terms of experimental method but it really needs to be explained better.

**Response:** First, under natural conditions, it has been observed that rills on the slope of spoil tips may be formed by multiple rainfall events or runoff from upslope. Previous works also showed that the rill network reached mature stage at the third rainfall event (Shen et al. 2020). Therefore, for better development of rill networks, each inflow rate experiment contained three successive scourings (i.e., 1st to 3rd scourings) with an interval of 24 h, respectively. Second, the pre-examination results showed that covering the slope with plastic film after each experiment could effectively prevent excessive soil moisture evaporation of the slope, and the soil moisture of each slope was generally the same after 24 h of resting. Therefore, the interval between each experiment was 24 h so as to ensure that the soil moisture of each experiment was basically consistent. We have added additional information to the experimental design section of the revised manuscript.

- Shen H O, Zheng F L, Zhang X C J, Qin, C (2020) Rill network development on

loessial hillslopes in China. Earth Surface Processes and Landforms, 45, 3178-3184. https:// 10.1002/esp.4958.

**Q6.** -Line 106-108. What is the relevance of this soil? It seems like it was something that was available, not something that was of interest to the regions? Is this an important regional soil or just something available? Please explain.

**Response:** The experimental station is located in the Guanzhong Plain of the Loess Plateau of China (Fig 1a), where clay loam is the dominant soil of the region (Ji et al.2020, Wu et al. 2021). Therefore, clay loam was used to simulate soil erosion of the spoil tips, and the research results are representative of the soil properties in the area.

- Wu L J, Wang W L, Kang H L, Zhao M, Guo M M, Bai Y, Su H, Nie H Y (2021) Differences in hydraulic erosion processes of the earth and earth-rock Lou soil engineering accumulation in the Loess Region. Chinese Journal of Applied Ecology, 31 (5):1587-1598. https:// 10.13287/j.1001-9332.202005.013.

- Wu J, Qi Y B, Chang Q R, Liu M Y, Bai L M (2021) Attribution of Lou Soil in Chinese Soil Taxonomy and Establishment of Representative Soil Series in Guanzhong Area. Acta Pedologica Sinica, 58 (2):357-371. https:// 10.11766/trxb201906240325.

**Q7.** -Section 2. Great experimental setup! Its impressive! I really liked how you compacted the soils. This is a world class setup.

**Response:** Thank you for your appreciation of our experimental study.

**Q8.** -Section 2.4.1. While I can understand why you are doing these calculations, they are not really fully utilised or useful without seeing and understanding rill cross-sections. This is needed to be included. Its seems that they are included without being of great use.

**Response:** Yes, we agree with your suggestion and have added the section about rill cross-sections (Section 3.3.3). Thank you again.

**Q9.** -Line 218. What's the timing between runs? Was any new material added or dis you just start from the previous surface? Was there a crust, armour?

**Response:** As described in question 5, the interval between each experiment was 24 h. In order to simulate the rill development, the next experiment was conducted on the

basis of the previous one and no new material was added to the slope. After each experiment, the slope was covered with plastic film and left for 24 h before the next experiment. We found that the slope surface occasionally showed a slight crusting phenomenon when the next experiment was conducted. The intensity of crust formation was inversely related to the slope, and generally soil crusting was rarely produced on steep slopes. This is because rill erosion often occurs on slope under steep slope conditions, which causes continuous erosion of the soil surface. In addition, once the flow path on the slope is formed under the conditions of scouring, the runoff will basically erode the soil along the fixed flow path. Therefore, we think that the occasional slight crusting phenomenon has minimal impact on the experiment.

**Q10.** - Section 3.1 is quite difficult to follow as the first paragraph discusses runoff rates and their variability and then jumps to rill growth then jumps to equations of runoff rate (RR). I don't follow why you have fitted equations for RR as I struggle to see where and how its used later? Also, I don't understand what N (scouring time) is? Is this start of the rill incision? Rill growth? Data on rate of rill growth would be interesting and useful.

**Response:** Slope runoff is one of the driving forces for stripping soil and is also a carrier of sediment transport. As the surface micro-topography causes the differential erosion of the overland flow, the runoff can be further gathered into rill flow. The slope surface begins to form step-down floor under the action of rill flow, which also marks the beginning of rill erosion. In the process of rill formation and development, the slope surface runoff and sediment production process also changed. On the one hand, the rill formation provides a channel for runoff and erosion products, and the runoff changes from surface runoff to rill runoff, which causes a sharp increase in erosion volume. On the other hand, the evolution of rill morphology affects the structure of runoff within the channel, thus affecting the runoff, infiltration, sediment transport and confluence in the process of slope erosion. In addition, the development process of rill morphology is finally presented by the change process of runoff rate and soil loss rate. In summary, the first observation of the occurrence of slope erosion is runoff and sediment, and the trace left on the slope after the flow eroded was the erosion rill. Therefore, we first show

the process of variation of runoff rate and soil loss rate. The final result of the slope soil erosion is presented as rill network, so we next analyzed the rill network and morphological characteristics changes and the relationship between rill morphological characteristics (e.g., rill width, rill depth and rill width to depth ratio) and erosion based on the analysis of runoff production and sediment production. Finally, we analyzed the hydraulic characteristics of rill flow and the dynamic mechanisms of rill erosion.

N is scouring times (i.e., 1st to 3rd scourings). In addition, the data on the growth of the rill we expressed mainly by the rill width, rill depth and rill width–depth ratio.

**Q11.** -Line 248. 'number of scouring'?

**Response:** We are so sorry for our unclear expression. What we want to express is the scouring times (i.e., 1st to 3rd scourings).

**Q12.** - Equation 9. Is this the best fit for all slopes? While interesting, this should be scaled for area and compared with other studies.

**Response:** In any field experiments of this nature, one has to select some typical slopes that best represent the real-world situation and in our case the choice of the three slopes (28°, 32° and 36°) was made based on survey of real-world spoil tips. Therefore, we believe that Equation 9 can be applied to such slope range. However, whether the equation is applicable to other slope ranges needs to be further studied.

We are really sorry that we did not pay attention to scaling area, when established the formula. Thank you for raising this problem. In this paper we focus on the effects of inflow rate (1.6, 2 and 2.4 mm min$^{-1}$), slope (28°,32° and 36°), and scouring times (1st to 3rd scourings) on rill erosion of the spoil tips. In addition, the soil loss rate is in units of g m$^{-2}$ min$^{-1}$, in the sense that we have included the area. Our future research will definitely pay special attention to the scaling. In addition, we add comparison with other studies in section 3.2.

**Q13.** -Figure 5. The 2mm runoff has quite a bit of scatter and in some cases more than the 2.4mm. Can you suggest why?

**Response:** The variation of soil loss rate is related to the rill morphology development (i.e., headward erosion, bank landslip erosion, and downcutting erosion). In the process of erosion, the rill interconnection erosion intensified, and the side walls on both sides

of the rill began to collapse. With the blocking and scouring of the side walls, the erosion and collapse occurred repeatedly, and erosion fluctuates, so that multiple peaks and lows occur during the erosion process. Under the condition of maximum inflow rate (2.4 mm min$^{-1}$), rill was basically developed after the first scouring experiment, and the soil loss rate in general fluctuated the most. Therefore, the soil loss rate fluctuation variation was smaller in the second and third experiments than in the first one. However, the rill development was slower with 2 mm min$^{-1}$ than 2.4 mm min$^{-1}$. With the increase of scouring times, rill bank landslip occurs, which leads to the soil loss rate of 2 mm min$^{-1}$ has quite a bit of scatter and in some cases more than the 2.4 mm min$^{-1}$.

**Q14.** - Figure 14. Interesting that rills developed at the top of the slope for the 32degree slope and not the others. Why? Would be very helpful to show some cross-sections.

**Response:** In the runoff scouring process, the flow and erosion forces are greatest when the water enters the slope from the top of the slope, thus the rill head appears at the top of the slope, and once the step-down floor can begin to appear and develop into a rill at the top of the slope, rill erosion will rapidly undergo headwater erosion, undercutting erosion and lateral erosion. As a result, the top of the slopes all show varying degrees of rill erosion (Fig.2). The rill cross-sections also indicate that rill erosion is more severe at the top of the 32° slope than at 28° under maximum inflow rate conditions (2.4 mm min$^{-1}$). For the 36° slope, although the slope is greater than 32°, the runoff erosion force is relatively weak due to the small inflow rate (2 mm min$^{-1}$). As a result, rills developed at the top of the slope for the 32°.

[Figure]

[Figure]

[Figure]

Figure 2. Rill cross-sections

**Q15.** -Figure 8 and accompanying text. I really struggled to see what this is demonstrating and ultimately where it is going.

**Response:** With figure 8 we mainly want to show the variation of rill characteristics (i.e., average rill width, average rill depth and rill width-to-depth ratio) with slope (28°, 32°and 36°), inflow rate (1.6, 2and 2.4 mm min$^{-1}$) and scouring times (1st to 3rd scourings). Figure 8 and accompanying text have been modified to better show the variation of the rill characteristics.

[Figure]

[Figure]

Figure 8. Variations in the mean rill width, mean rill depth and rill width-depth ratio with inflow rate (1.6, 2 and 2.4 mm min$^{-1}$), scouring times (1st to 3rd scourings) and slope (28°,32° and 36°).

**Q16.** -Equations 10,11 and 12. Some great work here but how does this compare to what other have found and for other soils?

**Response:** We thank you for your recognition of our study. We add to the discussion section in 4.2 a comparison with others' studies.

**Q17.** -I struggled to put Sections 3.4.1 onwards into context.

**Response:** Rill erosion is a non-linear dynamic process which is stochastic and complex. The rill erosion process consists of multiple sub-processes, with the rill morphology and sediment transport capacity of each process are largely influenced by

the hydrodynamic characteristics of the rill flow (An et al., 2012; Reichert and Norton, 2013). The runoff hydrodynamic characteristics represent the change in runoff erosion energy during rill erosion and reflect the ability of runoff to strip and transport soil particles on the slope (Mirzaee and Ghorbani-Dashtaki, 2018). Therefore, the rill flow hydrodynamic characteristics during rill erosion need to be studied. By analyzing the rill flow dynamic characteristics and erosion dynamics mechanism under different experiments conditions and identifying the best hydrodynamic parameters, it helps to fully understand the action mechanism of rill development on slope and provides a theoretical basis for the establishment of slope erosion prediction models. In addition, Figures 10 and 12 have been modified to better show the variation of the rill hydrodynamic parameters.

[Figure]

[Figure]

Figure 10. Variations in the mean flow velocity, Reynolds number, Froude number and Darcy-Weisbach coefficient with inflow rate (1.6, 2 and 2.4 mm min$^{-1}$), scouring times (1st to 3rd scourings) and slope (28°,32° and 36°).

[Figure]

Figure 12. Variations in the runoff shear stress, stream power and unit stream power with inflow rate (1.6, 2 and 2.4 mm min$^{-1}$), scouring times (1st to 3rd scourings) and slope (28°,32° and 36°).

- An J, Zheng F L, Lu J, Li G F (2012) Investigating the role of raindrop impact on hydrodynamic mechanism of soil erosion under simulated rainfall conditions. Soil Science. 177, 517–526. https:// 10.1097/SS.0b013e3182639de1.

- Reichert J M, Norton L D, Favaretto N, Huang C H, Blume E (2007) Settling velocity, aggregate stability, and inter-rill erodibility of soils varying in clay mineralogy. Soil Science Society of America Journal.73(4):1369-1377. https://10.2136/sssaj2007.0067.

- Mirzaee S, Ghorbani-Dashtaki S (2018) Deriving and evaluating hydraulics and detachment models of rill erosion for some calcareous soils. Catena. 164: 107-115. https:// 10.1016/j.catena.2018.01.016.